# HIV gp120 Protein Increases the Function of Connexin 43 Hemichannels and Pannexin-1 Channels in Astrocytes: Repercussions on Astroglial Function

**DOI:** 10.3390/ijms21072503

**Published:** 2020-04-03

**Authors:** Rosario Gajardo-Gómez, Cristian A. Santibañez, Valeria C. Labra, Gonzalo I. Gómez, Eliseo A. Eugenin, Juan A. Orellana

**Affiliations:** 1Departamento de Neurología, Escuela de Medicina and Centro Interdisciplinario de Neurociencias, Facultad de Medicina, Pontificia Universidad Católica de Chile, Santiago 8330024, Chile; rgajardo1@uc.cl (R.G.-G.); c.santibanezahumada@gmail.com (C.A.S.); valelabra@gmail.com (V.C.L.); 2Institute of Biomedical Sciences, Faculty of Health Sciences, Universidad Autónoma de Chile, Santiago 8910060, Chile; gonzalo.gomez@uautonoma.cl; 3Department of Neuroscience, Cell Biology, and Anatomy, University of Texas Medical Branch, (UTMB), Galveston, TX 77555, USA; eleugeni@utmb.edu

**Keywords:** Cx43 hemichannels, connexins, gp120, glia, HIV, astrocyte

## Abstract

At least half of human immunodeficiency virus (HIV)-infected individuals suffer from a wide range of cognitive, behavioral and motor deficits, collectively known as HIV-associated neurocognitive disorders (HAND). The molecular mechanisms that amplify damage within the brain of HIV-infected individuals are unknown. Recently, we described that HIV augments the opening of connexin-43 (Cx43) hemichannels in cultured human astrocytes, which result in the collapse of neuronal processes. Whether HIV soluble viral proteins such as gp120, can regulate hemichannel opening in astrocytes is still ignored. These channels communicate the cytosol with the extracellular space during pathological conditions. We found that gp120 enhances the function of both Cx43 hemichannels and pannexin-1 channels in mouse cortical astrocytes. These effects depended on the activation of IL-1β/TNF-α, p38 MAP kinase, iNOS, cytoplasmic Ca^2+^ and purinergic signaling. The gp120-induced channel opening resulted in alterations in Ca^2+^ dynamics, nitric oxide production and ATP release. Although the channel opening evoked by gp120 in astrocytes was reproduced in ex vivo brain preparations, these responses were heterogeneous depending on the CA1 region analyzed. We speculate that soluble gp120-induced activation of astroglial Cx43 hemichannels and pannexin-1 channels could be crucial for the pathogenesis of HAND.

## 1. Introduction

After primary peripheral infection, the human immunodeficiency virus (HIV) crosses the blood–brain barrier (BBB) using monocytes as a vehicle, gaining access to the central nervous system (CNS) [1,2]. After that, the HIV infects both microglia and a small population of astrocytes [3,4], adversely affecting proper glial cell function and concomitant survival of neighboring neurons [5,6]. The latter has been linked to the release of proinflammatory cytokines and free radicals, cytoplasmic Ca^2+^ imbalance and glutamate excitotoxicity [5,6]. In parallel, infected glial cells release soluble viral proteins into the brain parenchyma, triggering direct neuronal toxicity [7,8]. Overall these mechanisms are thought to contribute to a wide range of cognitive, behavioral and motor deficits, collectively known as HIV-associated neurocognitive disorders (HAND) [9,10]. Despite that antiretroviral therapy has diminished the severity of HIV cognitive disorders, the prevalence of HAND in HIV-infected patients remains high (50%) as they live longer and due to the relatively poor BBB penetrance of most antiretroviral drugs [11,12,13,14]. The prominent CNS damage observed in HIV-infected individuals with effective antiretroviral therapy had led to the thought that additional and novel mechanisms of bystander cell death might be implicated.

Embedded in the synaptic cleft, astrocytes govern crucial brain processes that include synaptic function and plasticity, energy supply for neurons, cytoplasmic Ca^2+^ signaling and homeostatic equilibrium of extracellular pH, neurotransmitters and ions, along with controlling the redox and inflammatory response [15,16,17,18]. Previous studies have shown that HIV itself or the HIV gp120, the surface glycoprotein responsible for viral entry, impairs astroglial function with particularly detrimental consequences for neuronal survival [19,20]. Indeed, in astrocytes HIV gp120 alters the exchange of Na^+^/H^+^ and glutamate efflux [21,22]; the levels of cytoplasmic Ca^2+^ [23] and the production of proinflammatory cytokines/chemokines and free radicals [24,25,26]. Despite the continuous ongoing research in the field, currently, there is no effective treatment for HAND and the precise underlying mechanism behind the pathological action of HIV and/or HIV gp120 remains not well understood.

A growing body of evidence has pointed out that cellular signaling mediated by hemichannels and pannexons might contribute to the dysfunction of astrocytes, with potentially significant repercussions for neuronal function and survival [27]. Hemichannels are plasma membrane channels containing six connexin subunits that oligomerize around a central pore, allowing autocrine/paracrine signaling via the exchange of ions and small molecules between the cytoplasm and the extracellular space [28]. Pannexins channels—also called pannexons—result from the oligomerization of pannexins, a three-member family of proteins with similar secondary and tertiary structures than connexins that establish plasma membrane channels permeable to ions and small molecules [29]. In pathological scenarios, instead of being beneficial, the persistent function of hemichannels and pannexons contributes to cell damage and dysfunction through different mechanisms, such as the release of potentially toxic molecules, intracellular Ca^2+^ imbalance and transmembrane ionic/osmotic disturbances [30,31]. In a prior study, we showed that HIV increases the opening of hemichannels but not pannexin channels in cultured human astrocytes, which result in the collapse of neuronal processes [32]. However, whether HIV derived viral proteins, including gp120, regulate hemichannel and/or pannexon opening in astrocytes is still unknown.

## 2. Results

### 2.1. HIV gp120 Protein Increases the Function of Cx43 Hemichannels and Panx1 Channels in Cultured Astrocytes

Given that the entire HIV virion induces the opening of Cx43 hemichannels in astrocytes [32], we examined whether the HIV envelope glycoprotein gp120 could affect the function of these channels in primary cortical astrocytes. The functional state of hemichannels was studied by recording the rate of ethidium (Etd) uptake. This fluorescent probe crosses the plasma cell membrane by diffusing through channels with large pores such as hemichannels [33]. In these experiments, gp120 caused a significant bell-shaped increase in astrocyte Etd uptake that peaked a 2.3-fold increase following 24 h of treatment and gradually declined over the days (Figure 1A–E). Furthermore, the stimulus for 24 h with gp120 was also bell-shaped depending on its concentration and reaching the highest value with 10 ng/mL treatment (Figure 1A,B). Thus, this concentration and time of treatment were used in all further experiments.

Since Cx43 hemichannels and Panx1 channels represent one of the most prevalent routes for dye influx in astrocytes [34,35], the potential contribution of these channels in the gp120-induced astroglial Etd uptake was examined. Accordingly, astrocyte cultures were preincubated for 15 min before and throughout Etd uptake recordings with various pharmacological agents. Tat-L2 (100 µM) or gap19 (100 µM); two inhibitory mimetic peptides with sequences equivalent to intracellular L2 loop regions of Cx43 [36,37,38]; completely blunted the gp120-induced Etd uptake in astrocytes to control values (Figure 1E,F). On the contrary, a mutated Tat-L2 (Tat-L2^H126K/I130N^), in which two amino acids essential for the interaction of the L2 region with the carboxyl tail of Cx43 were modified, did not evoke a similar inhibitory response (Figure 1F). An equivalent unsuccessful suppression was observed with an inactive form of gap19 containing the I130A variation (gap19^I130A^; Figure 1F). To further explore the implication of Panx1 channels in the gp120-induced Etd uptake in astrocytes, we used the mimetic peptide ^10^panx1 with an amino acid sequence complementary to the first extracellular loop region of Panx1 [39] and probenecid, a potent inhibitor of these channels [40]. Both ^10^panx1 (100 µM) and probenecid (500 µM) elicited a substantial inhibitory influence on the Etd uptake triggered by gp120 in astrocytes (Figure 1F). Overall, these results reveal that gp120 boosts the function of Cx43 hemichannels and Panx1 channels in cultured astrocytes.

### 2.2. gp120-Induced Hemichannel and Pannexon Function is Mediated by the Production of IL-1β/TNF-α and the Activation of p38 MAPK/iNOS/[Ca^2 +^]_i_/P2X_7_/P2Y_1_-Dependent Pathways

Prior evidence has described that the activation of astroglial hemichannels and pannexons during pathological conditions involves TNF-α/IL-1β, p38 MAPK, inducible NO synthase (iNOS), P2X_7_/P2Y_1_ receptors and cytoplasmic Ca^2+^ [41,42,43,44]. Consequently, we scrutinized the impact of these pathways in the gp120-induced Cx43 hemichannel and Panx1 channel function in astrocytes. The stimulation with sTNF-aR1 or IL-1ra, a soluble form of TNF-α receptor that binds TNF-or a recombinant antagonist for the IL-1β receptor, respectively, totally blunted the Etd uptake produced by 24 h of treatment with gp120 (Figure 2). Of note, the gp120-dependent Etd uptake was drastically suppressed by a blockade of p38 MAPK with 10 μM SB202190, whereas L-N6 (5 μM), an iNOS inhibitor, triggered a partial counteracting action (Figure 2). Given that rise in intracellular free Ca^2+^ concentration ([Ca^2+^]_i_) and purinergic signaling are broadly known mechanisms that increase the function of Cx43 hemichannels and Panx1 channels [42,45,46], we investigated if they were linked to the gp120-induced Etd uptake in astrocytes. Notably, chelation of [Ca^2+^]_i_ with 5 μM BAPTA-AM, 200 µM oATP (wide-spectrum P2X receptor blocker) or 200 nM A740003 (P2X_7_ receptor antagonist); induced a significant but partial reduction in the Etd uptake elicited by gp120 (Figure 2). In the same manner, neutralization of P2Y_1_ receptor activity with 10 µM MRS2179 blunted the gp120-induced Etd uptake (Figure 2). Our data indicate that the activation of Cx43 hemichannels and Panx1 channels induced by gp120 relies on the secretion of IL-1β/TNF-α and the activation of p38 MAPK/iNOS/[Ca^2+^]_i_/P2X_7_/P2Y_1_-dependent cascades.

### 2.3. gp120 did not Affect Astrocyte-to-Astrocyte Coupling or the Distribution of Cx43 in Astrocytes

Gap junctional-mediated coupling between astrocytes contributes to spatial buffering of K^+^, as well as the spread of energy substrates (e.g., lactate and glucose) and intercellular Ca^2+^ waves, these processes being critical to guarantee adequate neuronal function [47]. Relevantly, cell–cell uncoupling along with the increased opening of hemichannels and pannexons in astrocytes are concurrent phenomena that result in neuronal dysfunction and damage [41,43,44]. Keeping this into account, we evaluated whether gp120 alters the function of gap junction channels in astrocytes. Etd coupling experiments revealed that under control conditions, almost 100% of astrocytes were coupled (Figure 3A,B,E) and most of them did it with 14 neighboring astrocytes (Figure 3A,B,F). Surprisingly, 24 h of treatment with gp120 failed in alter astroglial coupling (Figure 3C–F) and the distribution of Cx43 in confluent astrocytes, the latter measured by immunofluorescence analysis (Figure 3G–J). Certainly, in both control and gp120-treated astrocytes, Cx43 was detected mostly and intensely as fine to large immunopositive spots scattered at cell–cell interfaces (Figure 3G–J). Altogether these findings indicate that gp120 specifically augments the function of Cx43 hemichannels but not the functional state of gap junction channels in astrocytes.

### 2.4. The gp120-Induced Release of ATP Depends on the Opening of Panx1 Channels but not Cx43 Hemichannels in Astrocytes

Neuropathological scenarios often are accompanied by the release of ATP, which acts as a pleiotropic danger signal, triggering reactive astrogliosis and engagement of microglia and other peripheral immune cells that augment the susceptibility of neurons to damage [48]. Since Cx43 hemichannels and Panx1 channels allow the efflux of ATP in astrocytes [35,49] and given that P2X_7_ and P2Y_1_ receptors take part in hemichannel/pannexon function elicited by gp120 (Figure 2), we further studied the release of this molecule. Incubation with gp120 for 24 h greatly raised the release of ATP by 3.7-folds in relation to untreated astrocytes (Figure 4). Both Tat-L2 or gap19 were ineffective in preventing this response, whereas ^10^panx1 or probenecid fully counteracted the release of ATP elicited by gp120 (Figure 4). Collectively, this evidence implies that gp120 triggers the efflux of ATP in astrocytes by a mechanism that needs the opening of Panx1 channels but not Cx43 hemichannels.

### 2.5. The gp120-Induced Production of NO is Partially Mediated by the Activation of Cx43 Hemichannels and Panx1 Channels in Astrocytes

The production of IL-1β and TNF-α has been proposed as a key element not only in the dysfunction of astrocytes [50], but also in the uncontrolled function of astroglial Cx43 hemichannels in pathological circumstances [41,51]. Considering this, along with the fact that sTNF-aR1 and IL-1ra dramatically counteracted the gp120-induced Etd uptake by astrocytes (Figure 2), we tested whether this viral protein could perturb the release of IL-1β and TNF-α in our system. Astrocytes stimulated with gp120 for 24 h displayed a 4.5-fold and 6-fold rise in the secretion of IL-1β and TNF-α compared to untreated conditions, respectively (Figure 5A,B). Contrary to prior evidence indicating that the activation of hemichannels and pannexons contributes to the production of cytokines [39,52,53,54], we observed that Tat-L2, gap19 or ^10^panx1 did not reduce the gp120-induced release of IL-1β and TNF-α. Increased iNOS activation and subsequent production of NO are downstream processes of IL-1β/TNF-α signaling and participate in reactive astrogliosis [55]. With this in mind and because LN-6, a specific iNOS blocker, significantly inhibited the gp120-induced Etd uptake in astrocytes (Figure 2), we evaluated whether Cx43 hemichannels or Panx1 channels disturbs NO production. Recordings of DAF-FM fluorescence signal showed that treatment with gp120 triggered a 3.3-fold increase in basal NO production in relation to control conditions (Figure 5C–E). Noteworthy, gap19 or ^10^panx1 strongly abolished this response (Figure 5C,F,G), unveiling that both Cx43 hemichannels and Panx1 channels were fundamental.

### 2.6. Cx43 Hemichannels and Panx1 Channels Participate in the gp120-Induced Increase in ATP-Mediated [Ca^2+^]_i_ Dynamics by Astrocytes

Changes in astroglial [Ca^2+^]_i_ dynamics and the opening of Cx43 hemichannels are mutual processes that occur in neuroinflammatory scenarios [42,56,57]. Given this and because intracellular BAPTA significantly blunted gp120-dependent Etd uptake (Figure 2), we examined if gp120 could modulate the basal Ca^2+^ signal in astrocytes. Recordings of Fura-2 ratio (340/380) showed that gp120 did not produce relevant disturbances in basal Ca^2+^ levels in relation to control conditions (Figure 6A,C,E,F,I). Nonetheless, the above does not rule out whether gp120 could alter Ca^2+^ dynamics evoked by autocrine/paracrine signals. Since extracellular levels of ATP were found increased in astrocytes treated with gp120 (Figure 4), we studied the impact of this viral protein on ATP-mediated Ca^2+^ dynamics. As previously shown [44], 500 µM ATP induced a quick Ca^2+^ signal response with a slight amplitude in control astrocytes (Figure 6B,E,J). However, gp120 caused a prominent ATP-mediated Ca^2+^ signal with an amplitude 2-fold bigger than control values (Figure 6D,F,J). This response took place in parallel with a 2.5-fold and 2-fold rise in the integrated area under curve of ATP-dependent Ca^2+^response (Figure 6K) and the residual difference between the final and initial basal Ca^2+^ signal (Figure 6L), respectively.

It has been proposed that both hemichannels and pannexons are conduits for Ca^2+^ [58,59] and regulate cytoplasmic Ca^2+^ dynamics through the efflux of signals (e.g., ATP) [46]. Consistent with this idea, we noted that the blockade of Cx43 hemichannels or Panx1 channels with gap19 or ^10^panx1 strongly diminished the increase in ATP-mediated Ca^2+^ signal amplitude triggered by gp120 (Figure 6G,H,J). In a similar way, the blockade of Cx43 hemichannels and Panx1 channels prominently abolished the augment in the area under the curve and remaining basal ATP-dependent Ca^2+^ responses induced by gp120 (Figure 6K,L). On the whole, these data point out that the function of Cx43 hemichannels and Panx1 channels participate in the gp120-induced rise of ATP-mediated Ca^2+^ dynamics in astrocytes.

### 2.7. gp120 Elevates the Function of Astroglial Cx43 Hemichannels and Panx1 Channels in the Hippocampus

To scrutinize the impact of gp120 on astrocytes embraced in a more physiological context, we explored if this viral protein could modify the function of hemichannels and pannexons in hippocampal astrocytes from acute brain slices. Etd uptake by GFAP-positive astrocytes was examined in three areas of the hippocampus: the stratum oriens, stratum pyramidale and stratum radiatum. In control brain slices, astrocytes displayed a slight Etd uptake in all CA1 studied areas (Figure 7A, Figure 8A and Figure 9A). Nevertheless, following 30 min of gp120 stimulation, the hippocampus exhibited astrocytes with enhanced Etd uptake at the stratum oriens (11.3-fold, Figure 7B–C), stratum pyramidale (8-fold, Figure 8B–C) and stratum radiatum (12.5-fold, Figure 9B–C). Analysis of the temporal responses uncovered that Etd uptake quickly rose following 30 min of gp120 treatment but gradually declined over the hours (Figure 7A, Figure 8A and Figure 9A). Further, acute brain slices were preincubated for 15 min before and during Etd uptake experiments with inhibitors for Cx43 hemichannels or Panx1 channels in order to inspect their contribution in these responses. Equivalent to what we observed in astrocyte cultures (Figure 1F), 100 µM ^10^panx1 significantly abolished the gp120-induced Etd uptake observed in astrocytes from the stratum oriens (45%, Figure 7D), stratum pyramidale (46%, Figure 8D) and stratum radiatum (36%, Figure 9D). Nevertheless, gap19 (100 µM) was successful in diminishing Etd uptake only in the stratum oriens (33%, Figure 7D) but not in the stratum pyramidale (Figure 8D) or stratum radiatum (Figure 9D). Taking together these findings show that gp120 augments the function of astroglial Panx1 channels in at distinct hippocampal regions, whereas its stimulatory effect on astroglial Cx43 hemichannels is restricted to the stratum oriens.

## 3. Discussion

Here, we demonstrated that gp120 increased the function of Cx43 hemichannels and Panx1 channels in astrocytes. This stimulatory effect took place via the signaling of IL-1β/TNF-α and the activation of p38 MAPK/iNOS/[Ca^2+^]_i_-dependent cascades and P2Y_1_/P2X_7_ purinergic receptors. Remarkably, the function of Cx43 hemichannels and Panx1 channels evoked by gp120 was crucial for inducing severe alterations in ATP release, NO production and [Ca^2+^]_i_ dynamics in astrocytes.

Time-lapse recordings of Etd uptake demonstrated that gp120 increased the function of Cx43 hemichannels and Panx1 channels in a time and concentration-dependent form in primary cortical astrocytes. Two well-recognized specific mimetic peptides that antagonize Cx43 hemichannel opening (Tat-L2 and gap19), drastically neutralized the gp120-induced Etd uptake. In addition, the inhibition of Panx1 channels with ^10^panx1 or probenecid resulted in similar antagonistic effects, revealing that both Cx43 hemichannels and Panx1 channels were significant protagonists in the gp120-induced Etd uptake. These data are consistent with prior in vitro and ex vivo studies describing the parallel opening of both channels in astrocytes subjected to neuroinflammatory conditions such as FGF-1 [60], alcohol [61], ultrafine carbon black [62], gp120 [44], familial Alzheimer’s disease [63], spinal cord injury [64] and acute infection [65].

The functional state of gap junction channels and hemichannels is inversely modulated in inflamed astrocytes [41,66]. In discrepancy with this evidence, we observed that gp120 did not reduce astroglial coupling, as recorded by intercellular Etd transfer. As inferred from confocal immunofluorescence studies, gp120 had no effects on the localization of Cx43 in astrocytes, suggesting that the organization of gap junctions remain unaltered. One hypothesis to explain this apparent contradiction is that gap junctions could keep their functional state to spread and amplify gp120-mediated toxic substances. The latter it has been demonstrated to occur between HIV-1-infected and uninfected astrocytes [20]. The variety of potentially toxic molecules diffusing through gap junction is wide, ranging from free radicals to high concentrations of cytoplasmic Ca^2+^, as well as IP_3_ [67,68,69,70].

Mounting evidence has shown that gp120 triggers the persistent activation of astrocytes associated with a broad-spectrum generation of inflammatory signals, including IL-1β and TNF-α [71,72]. Certainly, the expression of both cytokines augments in postmortem brains of HIV-1 patients [73] and their signaling linked to the p38 MAPK pathway opens astroglial Cx43 hemichannels [41,44,51,74]. By making use of a combination of selective inhibitors, we observed that gp120-induced Etd uptake embraces the activation of IL-1β/TNF-α and p38 MAPK, being this consistent with the fact that gp120 stimulates p38 MAPK in astrocytes [75]. Downstream signaling of IL-1β/TNF-α and p38 MAPK leads to the expression of iNOS [76] and, in consequence, increases the amounts of NO [77]. Despite that NO is essential for synaptic transmission and plasticity [78], its high production has been linked to astroglial-mediated neurotoxicity [79] and the opening of astroglial Cx43 hemichannels via NO-mediated S-nitrosylation of Cx43. Conformity with this, we found that gp120 dramatically augmented NO production in astrocytes, this effect being moderately abrogated by inhibition of Cx43 hemichannels or Panx1 channels. The latter harmonizes with previous studies describing that activation of Cx43 hemichannels/Panx1 channels and NO production are reciprocal processes occurring on glial cells exposed to proinflammatory conditions [44,80].

Prior data indicate that gp120 may induce the release of several “danger” signals from glial cells, including ATP [81], the latter molecule being recently proposed as a biomarker of HIV-1-mediated cognitive impairment [82]. With this in mind, two significant elements underscore the importance of ATP signaling on the gp120-induced astrocyte changes in our system. On the one hand, we found that suppression of both P2X_7_ and P2Y_1_ receptors greatly abrogated the Etd uptake evoked by gp120. At the other end, the activation of Panx1 channels was crucial for the release of ATP in gp120-stimulated astrocytes. Previous evidence has highlighted that ATP causes its release through hemichannels or pannexons, leading to the subsequent stimulation of purinergic receptors [44,74,83]. We conjecture that ATP release may serve as a downstream mechanism that results in the opening of Cx43 hemichannel and/or Panx1 channels, where P2Y_1_/P2X_7_ receptor-dependent rise in [Ca^2+^]_i_ could be fundamental, as already reported [44,74,83]. Indeed, a mild rise in [Ca^2+^]_i_ up to 500 nM significantly reinforces the function of Cx43 hemichannels [42], while analogous responses seem to occur in Panx1 channels [45]. In this line, we observed that BAPTA_,_ strongly reduced the gp120-induced Etd uptake in astrocytes. This is consistent with recent works showing that gp120 increases [Ca^2+^]_i_ in astrocytes [84] and that purinergic receptors are critical for HIV infection and gp120-mediated signaling [85,86]. Alongside this, given that Cx43 hemichannels are permeable to Ca^2+^ [56] and one could infer the same for Panx1 channels [58,87], they potentially may contribute to sustaining [Ca^2+^]_i_-dependent pathways linked to ATP release (see below).

Both the P2Y receptor-dependent release of internal Ca^2+^ and its extracellular influx via P2X receptors are emblematic astrocyte [Ca^2+^]_i_ responses evoked by ATP [88]. In this study, we observed that although gp120 had no consequences on basal levels of [Ca^2+^]_i_, it caused a drastic augment in ATP-induced Ca^2+^ responses, particularly, concerning the signal amplitude, integrated area under the curve and sustained signal. Worthy of note, the blockade of Cx43 hemichannels or Panx1 channels strongly antagonized the increased ATP-mediated [Ca^2+^]_i_ responses triggered by gp120. With this in mind, we speculated that the release of ATP and/or its derivates (e.g., ADP) from astrocytes might spread the gp120-mediated signaling to neighboring cells, resulting in Ca^2+^ responses that may impair the function and eventually the survival of glial cells and neurons. In such circumstances, the opening of Cx43 hemichannels and Panx1 channels could be crucial, whereas the signaling of purinergic receptors likely will be counteracted by 1) diffusion of ATP towards distant areas; 2) desensitization of P2Y_1_ receptors, 3) degradation of extracellular ATP via exonucleases and 4) self-inhibition of Panx1 channels by the direct action of ATP [89,90].

Although in vitro culture preparations are useful to dissect cellular mechanisms, they not always recapitulate the processes that take place in vivo. As a result of the use of acute brain slices, we corroborated in a much comprehensive model the stimulatory action of gp120 on Cx43 hemichannel/Panx1 channel function observed in astrocyte cultures. It is relevant to mention that gp120 exposure quickly augmented in a transient form the opening of Panx1 channels overall in the CA1 region, but only reproduce this effect on Cx43 hemichannel function in the stratum oriens. One plausible explanation is the existence of local stimuli (e.g., elevation in extracellular K^+^) affecting specifically Panx1 channels rather than Cx43 hemichannels in the CA1 region of the hippocampus. Alternatively, the differential influence of gp120 on channel opening may rely on the unique topological organization of hippocampal area CA1 astrocytes and pyramidal neurons [91]. For instance, astrocytes near to the stratum pyramidale are arranged in circuits that stay parallel to this layer, while astrocytes in the stratum radiatum constitute circular circuits [92].

Similarly, although the number of astrocytes remains unchanged between young and middle-aged mice, their quantity declines with the age in the stratum oriens, whereas the opposite occurs in the stratum lacunosum-moleculare [93]. In the same line, the electrophysiological features, cell–cell coupling, antigen profiles and [Ca^2+^]_i_ responses of astrocytes at the hippocampus are diverse depending on their location in this brain region [91,93,94,95]. The heterogeneity of neuronal types (e.g., pyramidal, baskets, etc.) and their projections (e.g., inhibitory and excitatory) is another factor that turns even more complex in these analyses. Likely, this diversity may account for the gp120-induced mediated heterogeneity in channel responses in the regions analyzed. Further research is required to unveil the mechanisms of these differential effects.

Overall, our findings support the idea that gp120-induced hemichannel/pannexon activation could occur rapidly upon HIV-1 brain invasion and being consequences of this process: i) the increase in ATP release, ii) the augment of NO production and iii) the rise in ATP-mediated [Ca^2+^]_i,_ dynamics. We propose a novel mechanism by which gp120 could disturb the astrocyte function implicating the successive activation of inflammatory cascades that in consequence enhance the activation of astroglial hemichannels and pannexons. The molecular mechanisms behind this phenomenon could serve as pharmacological targets for exploring new therapies aiming to tackle the pathogenesis and progression of HAND.

## 4. Materials and Methods

### 4.1. Reagents and Antibodies

Dulbecco’s Modified Eagle Medium (DMEM), water, L-N6, SB203580, MRS2179, oxidized ATP (oATP), HEPES, anti-GFAP monoclonal antibody, A74003, Cx43 rabbit polyclonal antibody (SAB4501174), ethidium (Etd) bromide and probenecid (Prob) were purchased from Sigma-Aldrich (St. Louis, MO, USA). Anti-Cx43 monoclonal antibody (610061) was obtained from BD Biosciences (Franklin Lakes, NJ, USA). Penicillin, BAPTA-AM, FURA-2AM, diamidino-2-phenylindole (DAPI), goat anti-mouse Alexa Fluor 488, streptomycin, DAF-FM diacetate, goat anti-mouse Alexa Fluor 488/555 and goat anti-rabbit Alexa Fluor 488/555 were from Thermo Fisher Scientific (Waltham, MA, USA). Fetal bovine serum (FBS) was purchased from Hyclone (Logan, UT, USA). Normal goat serum (NGS) was purchased from Zymed (San Francisco, CA, USA). A soluble form of the TNF-α receptor (sTNF-αR1) and a recombinant receptor antagonist for IL-1β (IL-1ra) were from R&D Systems (Minneapolis, MN, USA). The mimetic peptides gap19 (KQIEIKKFK, intracellular loop domain of Cx43), gap19^I130A^ (KQAEIKKFK, negative control), Tat-L2 (YGRKKRRQRRR-DGANVDMHLKQIEIKKFKYGIEEHGK, second intracellular loop domain of Cx43), Tat-L2^H126K/I130N^ (YGRKKRRQRRR-DGANVDMKLKQNEIKKFKYGIEEHGK, negative control) and ^10^panx1 (WRQAAFVDSY, first extracellular loop domain of Panx1) were obtained from Genscript (New Jersey, NJ, USA). The HIV-1 BaL gp120 recombinant protein recombinant protein (Cat#4961) was obtained from the NIH AIDS Reagent Program, Division of AIDS, NIAID and NIH.

### 4.2. Animals

C57BL/6 (PUC/The Jackson Laboratory) male mice of 2-month-old were housed in cages at temperature- (24 °C) and humidity-controlled ambient under a 12 h light/dark cycle (lights on 8:00 AM), with ad libitum access to food and water. Animal protocols were conducted following the guideline and approved protocol for care and use of experimental animals of the Bioethics Committee of the Pontificia Universidad Católica de Chile (PUC; n°: 150806013, 6 June 2016).

### 4.3. Cell Cultures

Primary cultures of cortical astrocytes were obtained from cortices of postnatal day 2 mice as previously reported [44]. After dissection of cortices, meninges were carefully peeled off and tissue was mechanically dissociated in Ca^2+^ and Mg^2+^ free Hank’s balanced salt solution (CM-HBSS) with 0.25% trypsin and 1% DNase. Cells were seeded onto 60-mm plastic dishes (Corning, NY, USA) or onto glass coverslips (Fisher Scientific, Waltham, MA, USA) placed inside 16-mm 24-well plastic plates (Corning, NY, USA) in DMEM, supplemented with streptomycin (5 µg/mL), penicillin (5 U/mL) and 10% FBS. Cells were grown at 37 °C in a 5% CO_2_/95% air atmosphere at nearly 100% relative humidity. Of cytosine-arabinoside 1 µM was added for 3 days after 8–10 days in vitro to eliminate proliferating microglia. Medium was changed twice a week and cultures were used after 3 weeks.

### 4.4. Treatments

Astrocytes were treated for 0, 1, 24, 48 or 72 h with 1, 5, 10 or 20 ng/mL of HIV-1 BaL gp120 (from now on referred as to “gp120”). To obtain conditioned media (CM) from astrocytes, cells (2 × 10^6^ cells in 35 mm dishes) were treated with 10 ng/mL gp120 for 24 h and supernatants filtered (0.22 µm), and stored at −20 °C before used for experiments. Different antagonists were preincubated 1 h prior and co-incubated with 10 ng/mL gp120 before experiments: mimetic peptides against Cx43 hemichannels (Tat-L2 and gap19, 100 µM) and Panx1 channels (^10^panx1, 100 µM), Prob (Panx1 channel blocker, 500 µM), sTNF-αR1 (soluble form of the receptor that binds TNF-α), IL-1ra (IL-1β receptor endogenous blocker), SB203580 (p38 MAP kinase inhibitor, 1 µM), L-N6 (iNOS inhibitor, 1 µM), BAPTA-AM (intracellular Ca^2+^ chelator, 10 µM), oATP (general P2X receptor blocker, 200 µM), MRS2179 (P2Y_1_ receptor blocker, 1 µM) and A740003 (P2X_7_ receptor blocker, 200 nM).

### 4.5. Dye Uptake and Time-lapse Fluorescence Imaging

Astrocytes plated on glass coverslips were washed twice in Hank’s balanced salt solution and bathed at room temperature with Locke’s solution (154 mM NaCl, 5.4 mM KCl, 2.3 mM CaCl_2_, 5 mM HEPES, pH 7.4) containing 5 µM Etd. Cells were then visualized with an Olympus BX 51W1I upright microscope with a 40× water immersion objective for time-lapse imaging. Images were captured using a Retiga 1300I fast-cooled monochromatic digital camera (12-bit; Qimaging, Burnaby, BC, Canada) controlled by imaging Metafluor software (Universal Imaging, Downingtown, PA) every 30 s (exposure time = 0.5 s; excitation and emission wavelengths were 528 nm 598 nm, respectively). The fluorescence intensity recorded from ≥30 regions of interest (representing at least 30 cells per cultured coverslip) was defined with the following formula: corrected total cell Etd fluorescence = integrated density – ((area of the selected cell) × (mean fluorescence of background readings)). The Etd uptake rate represent the mean slope of the relationship over a given time interval (ΔF/ΔT). To examine variations in the slope, regression lines were fitted using Microsoft Excel, and mean slope values were analyzed employing GraphPad Prism software and expressed as AU/min. In each independent experiment three replicates were performed. In some experiments, cultured astrocytes were preincubated with gap19 (100 µM), Tat-L2 (100 µM), ^10^panx1 (100 µM) or probenecid (500 µM) for 15 min before and during the Etd uptake.

### 4.6. Dye Coupling

Astrocytes plated on glass coverslips were iontophoretically microinjected with a glass micropipette filled with 75 mM Etd in recording medium (HCO_3-_-free F-12 medium buffered with 10 mM HEPES, pH 7.2) containing 200 μM La^3+^. This blocker was used to prevent cell leakage of the microinjected Etd via hemichannels, which could underestimate the transfer of Etd to neighboring cells. Astrocytes were visualized through a Nikon inverted microscope equipped with epifluorescence illumination (Xenon arc lamp) and Nikon B filter to Etd (excitation wavelength 528 nm; emission wavelength above 598 nm) and XF34 filter to DiI fluorescence (Omega Optical, Inc., Brattleboro, VT, USA). Photomicrographs were captured employing a CCD monochrome camera (CFW-1310M; Scion; Frederick, MD, USA). Five minutes after dye injection, the coupling incidence was calculated as the percentage of injections that resulted in Etd transfer from the injected cell to more than one neighboring cell, whereas the coupling index was calculated as the mean number of cells to which the Etd spread. Etd coupling was tested by microinjecting a minimum of 10 cells per experiment.

### 4.7. Immunofluorescence and Confocal Microscopy

Astrocytes plated on glass coverslips were with 2% paraformaldehyde (PFA) for 30 min fixed at room temperature. After washing three times with PBS, they were rinsed three times with (5 min each) 0.1 M PBS-glycine, and then incubated with 0.1% Triton X-100 in PBS containing 10% NGS for 30 min. Then astrocytes were incubated with an anti-GFAP monoclonal antibody (BD Biosciences, 1:400) or anti-Cx43 polyclonal antibody (SIGMA, 1:400) diluted in 0.1% Triton X-100 in PBS with 2% NGS at 4 °C overnight. After five rinses in 0.1% Triton X-100 in PBS, cells were incubated with goat anti-mouse IgG Alexa Fluor 355 (1:1000) or goat anti-rabbit IgG Alexa Fluor 488 (1:1000) at room temperature for 50 min. After washing, coverslips were mounted in DAKO fluorescent mounting medium and examined with an Olympus BX 51W1I upright microscope with a 40× water immersion. Nuclei were stained with DAPI or Hoechst 33342.

### 4.8. [Ca^2+^]_i_ and NO Imaging

Astrocytes plated on glass coverslips were loaded with 5 µM Fura-2-AM or 5 µM DAF-FM in DMEM without serum at 37 °C for 45 min and then washed three times in Locke’s solution followed by de-esterification at 37 °C for 15 min. The experimental protocol for [Ca^2+^]_i_ and nitric oxide (NO) imaging involved data acquisition every 5 s (emission at 510 and 515 nm, respectively) at 340/380-nm and 495 excitation wavelengths, respectively, using the same microscope and acquisition mentioned above for Etd uptake. The FURA-2AM ratio was obtained after dividing the 340-nm by the 380-nm fluorescence image on a pixel-by-pixel base (R = F_340 nm_/F_380 nm_).

### 4.9. Measurement of IL-1β, TNF-α and ATP Concentration

Extracellular amounts of IL-1β, TNF-α and ATP were measured in CM of astrocytes. Samples were centrifuged at 14.000× *g* for 40 min and then supernatants were collected and protein content analyzed through the bicinchoninic acid assay (BCA) technique. The amounts of IL-1β and TNF-α were determined by sandwich ELISA, as stated by the manufacturer (eBioscience, San Diego, CA, USA), whereas ATP levels were determined using a luciferin/luciferase bioluminescence assay kit (Sigma-Aldrich) as previously reported [44].

### 4.10. Acute Brain Slices

Coronal slices (300 µm) from anesthetized mice with isoflurane were obtained using a vibratome (Leica, VT1000GS; Leica, Wetzlar, Germany) in ice-cold slicing solution containing (in mM): sucrose (222); KCl (2.6); NaHCO_3_ (27); NaHPO_4_ (1.5); glucose (10); MgSO_4_ (7); CaCl_2_ (0.5) and ascorbate (0.1), bubbled with 95% O_2_/5% CO_2,_ pH 7.4. Then, the slices were transferred at room temperature (20–22 °C) to a holding chamber in ice-cold artificial cerebral spinal fluid (ACSF) containing (in mM): NaCl (125), KCl (2.5), glucose (25), NaHCO_3_ (25), NaH_2_PO_4_ (1.25), CaCl_2_ (2) and MgCl_2_ (1), bubbled with 95% O_2_/5% CO_2_, pH 7.4, for a stabilization period of 60 min before dye uptake experiments.

### 4.11. Dye Uptake in Acute Brain Slices and Confocal Microscopy

Acute brain slices were incubated with 25 µM Etd for 10 min in a chamber filled with ACSF and bubbled with 95% O_2_/5% CO_2_, pH 7.4. Afterward, the slices were washed three times (5 min each) with ACSF, and fixed at room temperature with 4% paraformaldehyde for 60 min, rinsed once with 0.1 mM glycine in phosphate-buffered saline (PBS) for 5 min and then twice with PBS for 10 min with gentle agitation. Then, the slices were incubated two times for 30 min each with a blocking solution (PBS, gelatin 0.2%, Triton-X 100 1%) at room temperature. Further, the slices were incubated overnight at 4 °C with an anti-GFAP monoclonal antibody (1:500, SIGMA) to detect astrocytes. Later, the slices were washed three times (10 min each) with a blocking solution and then incubated for 2 h at room temperature with goat anti-mouse Alexa Fluor 488 (1:1000) antibody and Hoechst 33342. Afterward, the slices were washed three times (10 min each) in PBS and then mounted in Fluoromount, cover-slipped and examined in a confocal laser-scanning microscope (Eclipse Ti-E C2, Nikon, Japan). Stacks of consecutive confocal images were taken with 40× objective at 100 nm intervals were acquired sequentially with three lasers (in nm: 408, 488 and 543), and Z projections were reconstructed using Nikon confocal software (NIS-elements) and ImageJ software. Etd uptake was calculated with the same formula mentioned for cell cultures. At least six cells per field were selected from at least three fields in each brain slice.

### 4.12. Data Analysis and Statistics

For each data group, results were expressed as mean ± standard error (SEM); *n* refers to the number of independent experiments. Detailed statistical results were included in the figure legends. Statistical analyses were performed using GraphPad Prism (version 7, GraphPad Software, La Jolla, CA, USA). Normality and equal variances were assessed by the Shapiro–Wilk normality test and Brown–Forsythe test, respectively. Unless otherwise stated, data that passed these tests were analyzed by unpaired *t*-test in case of comparing two groups, whereas in case of multiple comparisons, data were analyzed by a one or two-way analysis of variance (ANOVA) followed, in the case of the significance, by a Tukey’s post-hoc test. When data were heteroscedastic as well as not normal and with unequal variances, we used the Mann–Whitney test in the case of comparing two groups, whereas multiple comparisons data were analyzed by Kruskal–Wallis test followed, in the case of the significance, by a Dunn’s post-hoc test. A probability of *p* < 0.05 was considered statistically significant.

## Figures and Tables

**Figure 1 ijms-21-02503-f001:**
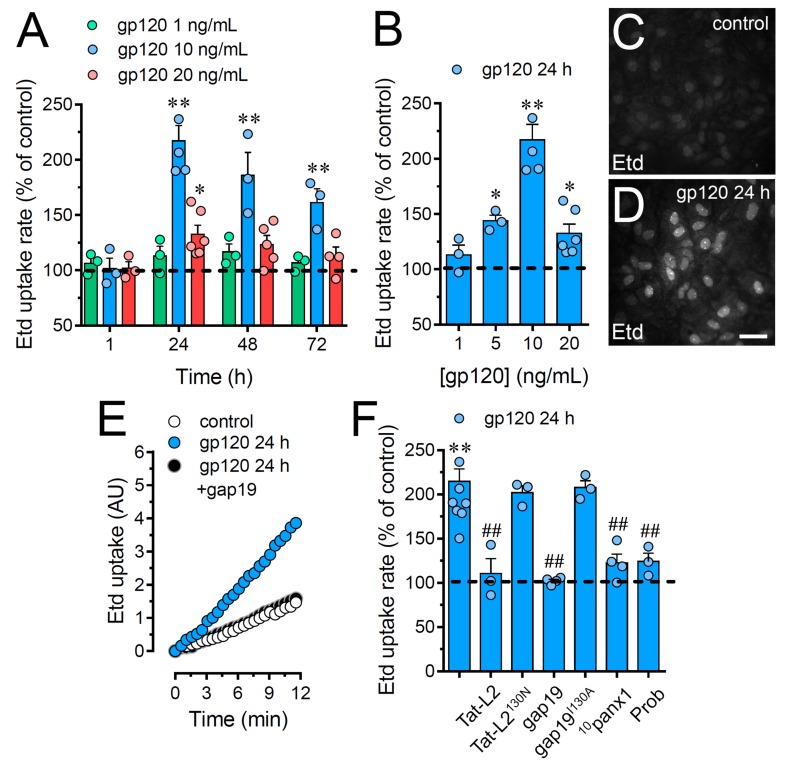
gp120 increases the function of Cx43 hemichannels and Panx1 channels in astrocytes. (**A**) Etd uptake rate normalized to control (dashed line) by astrocytes stimulated for distinct periods with gp120 at different concentrations: 1 ng/mL (green circles), 10 ng/mL (blue circles) or 20 ng/mL (red circles). * *p* < 0.05, ** *p* < 0.01, gp120 vs. control. (**B**) Etd uptake rate normalized to control (dashed line) by astrocytes stimulated for 24 h with distinct concentrations of gp120 (blue circles). * *p* < 0.05, ** *p* < 0.01, gp120 vs. control. (**C**,**D**) Etd staining from dye uptake measurements (10 min exposure to Etd) in astrocytes under control conditions (**C**) or stimulated for 24 h with 10 ng/mL gp120 (**D**). (**E**) Time-lapse recordings of Etd uptake by astrocytes under control conditions (white circles) or stimulated for 24 h with 10 ng/mL gp120 alone (blue circles) or plus 100 µM gap19 (black circles). (**F**) Etd uptake rate normalized to control (dashed line) by astrocytes stimulated for 24 h with 10 ng/mL gp120 alone or plus the following blockers: 100 µM Tat-L2, 100 µM Tat-L2^H126K/I130N^, 100 µM gap19, 100 µM gap19^I130A^, 100 µM ^10^panx1 or 500 µM Probenecid (Prob). ** *p* < 0.01, gp120 vs. control; # *p* < 0.05, ## *p* < 0.01;pharmacological agents vs. gp120. Data were obtained from at least three independent experiments (see scatter dot plot) with three or more repeats each one (≥ 30 cells analyzed for each repeat). Calibration bar = 45 μm.

**Figure 2 ijms-21-02503-f002:**
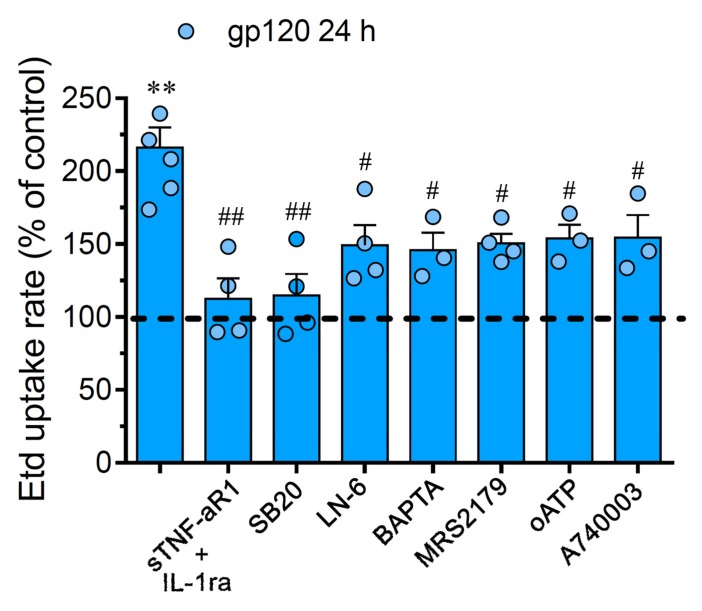
The gp120-induced Cx43 hemichannel and Panx1 channel function depends on IL-1β/TNF-α signaling and activation of p38 MAPK/iNOS/P2X_7_/P2Y_1_/[Ca^2+^]_i_-dependent pathways. Etd uptake rate normalized to control (dashed line) by astrocytes stimulated for 24 h with 10 ng/mL gp120 alone or plus the following agents: 100 ng/mL of IL-1ra+100 ng/mL of sTNF-αR1, 1 µM SB203580, 1 µM L-N6, 10 µM BAPTA, 1 µM MRS2179; 200 µM oxidized ATP (oATP) or 200 nM A740003. ** *p* < 0.01, gp120 vs. control; # *p* < 0.05, ## *p* < 0.01; pharmacological agents vs. gp120. Data were obtained from at least three independent experiments (see scatter dot plot) with three or more repeats each one (≥ 30 cells analyzed for each repeat).

**Figure 3 ijms-21-02503-f003:**
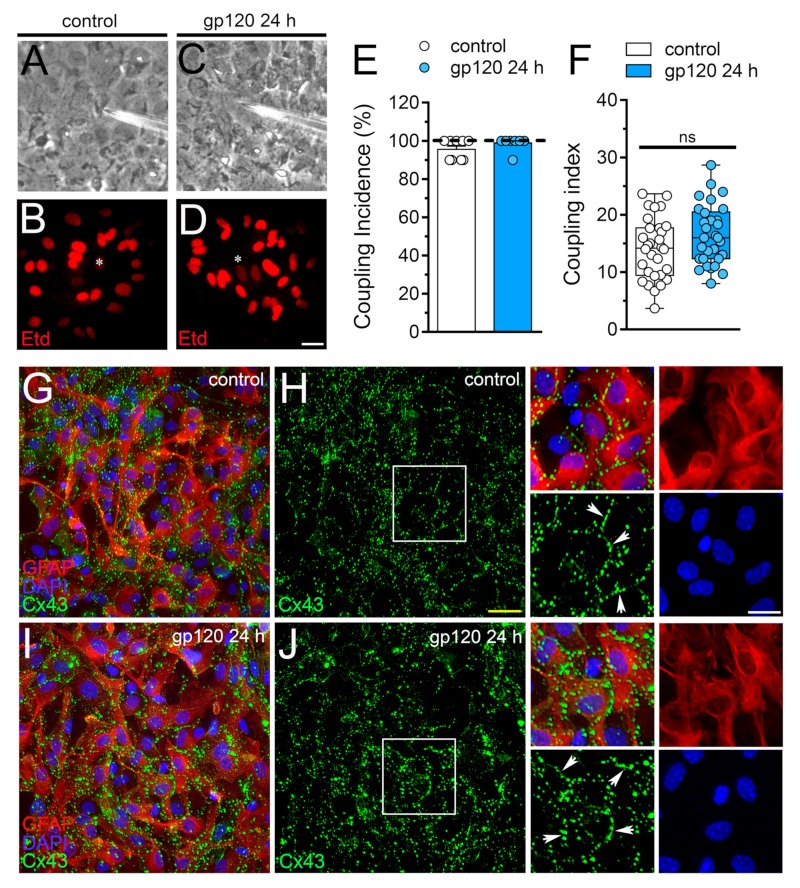
gp120 does not affect astroglial coupling or the distribution of Cx43 in astrocytes. (**A**–**D**) Fluorescence and phase-contrast micrographs of Etd transfer by astrocytes under control conditions (**A**,**B**) or stimulated for 24 h with 10 ng/mL gp120 (**C**,**D**). Yellow calibration bar = 60 µm. (**E**) Etd coupling incidence (percentage of injections that resulted in Etd transfer) of astrocytes under control conditions (white bar) or stimulated for 24 h with 10 ng/mL gp120 (blue bar). (**F**) Coupling index (number of cells coupled in positive injections) of astrocytes under control conditions (white bar) or stimulated for 24 h with 10 ng/mL gp120 (blue bar). White arrows indicate gap junction plaques at cell-cell interfaces. No significant differences were found. (**G**–**J**) Fluorescence images depicting Cx43 (green), GFAP (red) and DAPI (blue) staining by astrocytes under control conditions (**G**,**H**) or stimulated for 24 h with 10 ng/mL gp120 (**I**,**J**). Insets: 1.5× magnification of the white squares indicated area of panels H and J. Calibration bars: yellow = 120 µm and white= 15 µm.

**Figure 4 ijms-21-02503-f004:**
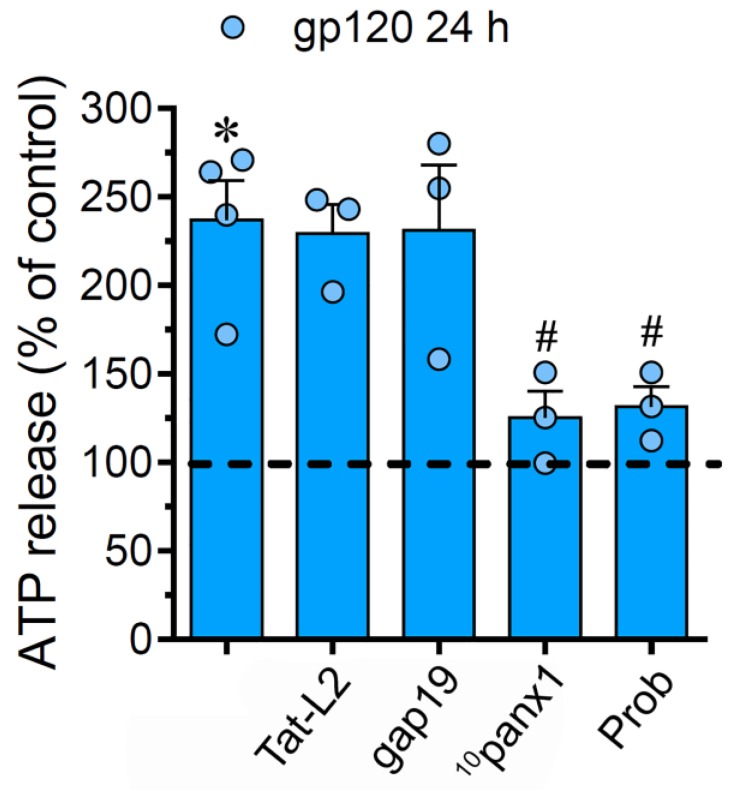
gp120 increases the release of ATP via the activation of Panx1 channels. ATP release normalized to the control (dashed line) by astrocytes stimulated for 24 h with 10 ng/mL gp120 alone or plus the following agents: 100 µM Tat-L2, 100 µM gap19, 100 µM ^10^panx1 and 500 µM Probenecid (Prob). * *p* < 0.01, gp120 vs. control; # *p* < 0.01, pharmacological agents vs. gp120. Data were obtained from at least three independent experiments (see scatter dot plot) with three or more repeats each one.

**Figure 5 ijms-21-02503-f005:**
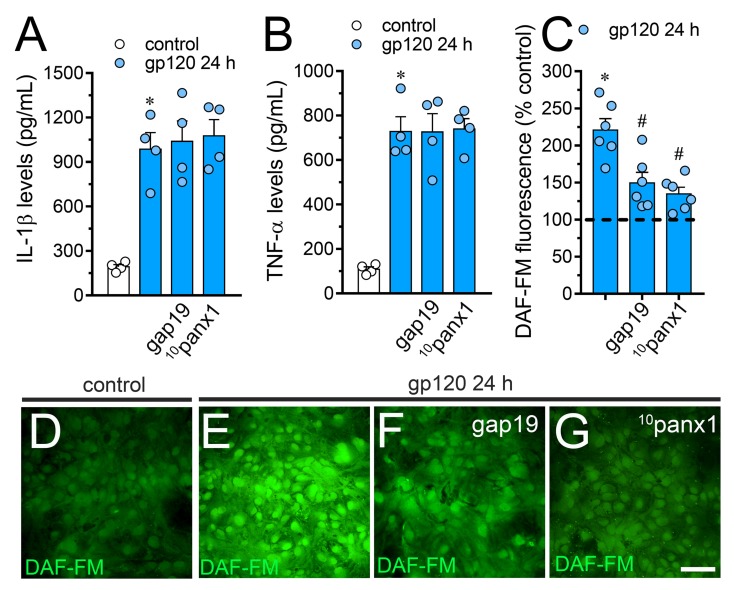
gp120 increases the release of IL-1β/TNF-α and the production of NO in astrocytes: involvement of Cx43 hemichannels and Panx1 channels. (**A**,**B**) Amounts of IL-1β (**A**) and TNF-α (**B**) found in the extracellular media of astrocytes cultured under control conditions (white bars) or stimulated for 24 h with 10 ng/mL gp120 alone (blue bars) or plus the following agents: 100 µM gap19 or 100 µM ^10^panx1. * *p* < 0.001, gp120 vs. control. Data were obtained from at least three independent experiments (see scatter dot plot) with three or more repeats each one. (**C**) DAF-FM fluorescence normalized to control (dashed line) by astrocytes stimulated for 24 h with 10 ng/mL gp120 alone or plus the following agents: 100 µM gap19 or 100 µM ^10^panx1. * *p* < 0.01, gp120 treatment vs. control, # *p* < 0.05, pharmacological agents vs. gp120. Data were obtained from at least three independent experiments (see scatter dot plot) with three or more repeats each one (≥ 35 cells analyzed for each repeat). (**D**–**G**) Fluorescence micrographs of basal NO production (DAF-FM, green) by astrocytes under control conditions (**D**) or stimulated for 24 h with 10 ng/mL gp120 alone (**E**) or plus 100 µM gap19 (**F**) or 100 µM ^10^panx1 (**G**). Calibration bars: white = 80 µm.

**Figure 6 ijms-21-02503-f006:**
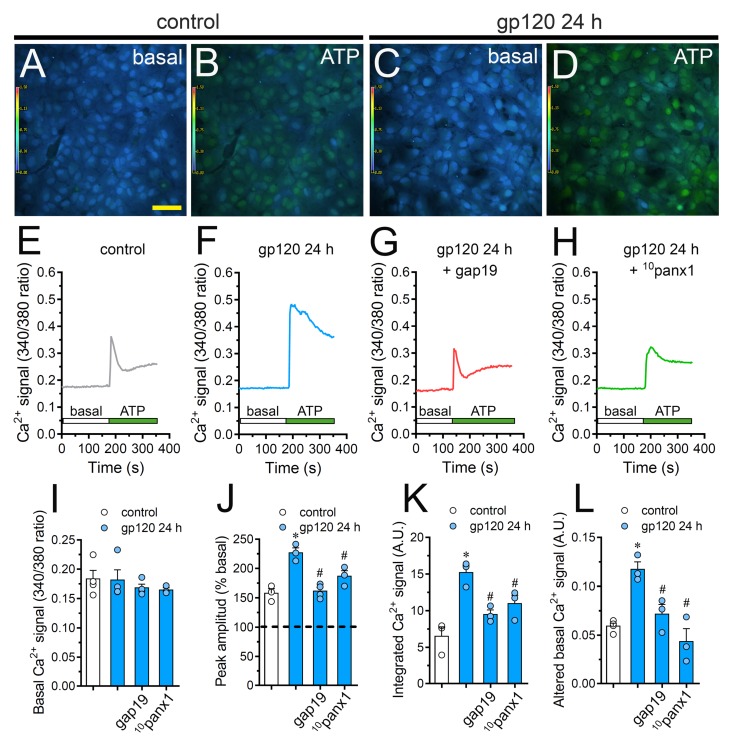
gp120 enhances ATP-dependent Ca^2+^ dynamics in astrocytes by a mechanism implicating the opening of Cx43 hemichannels and Panx1 channels. (**A**–**D**) Photomicrographs of basal (**A**,**C**) or 500 µM ATP-induced (**B**,**D**) Ca^2+^ signal denoted as Fura-2 ratio (340/380 nm excitation) of astrocytes under control conditions (**A**,**B**) or stimulated for 24 h with 10 ng/mL gp120 (C, D). Calibration bar: 60 μm. (**E**–**H**) Relative changes in Ca^2+^ signal over time induced by 500 µM ATP (green horizontal line) in astrocytes under control conditions (**E**) or stimulated for 24 h with 10 ng/mL gp120 (**F**) alone or plus the following agents: 100 µM gap19 (**G**) and 100 µM ^10^panx1 (**H**). (**I**–**L**) Basal Fura-2 ratio (I), ATP-induced peak amplitude normalized to basal Fura-2 ratio (**J**), integrated ATP-induced Fura-2 ratio response (**K**) and altered basal Fura-2 ratio (**L**) by astrocytes under control conditions (white bars) or stimulated for 24 h with 10 ng/mL gp120 (blue bars) alone or plus the following agents: 100 µM gap19 or 100 µM ^10^panx1. * *p* < 0.05, gp120 vs. control, # *p* < 0.05, pharmacological agents vs. gp120. Data were obtained from at least three independent experiments (see scatter dot plot) with three or more repeats each one (≥ 35 cells analyzed for each repeat).

**Figure 7 ijms-21-02503-f007:**
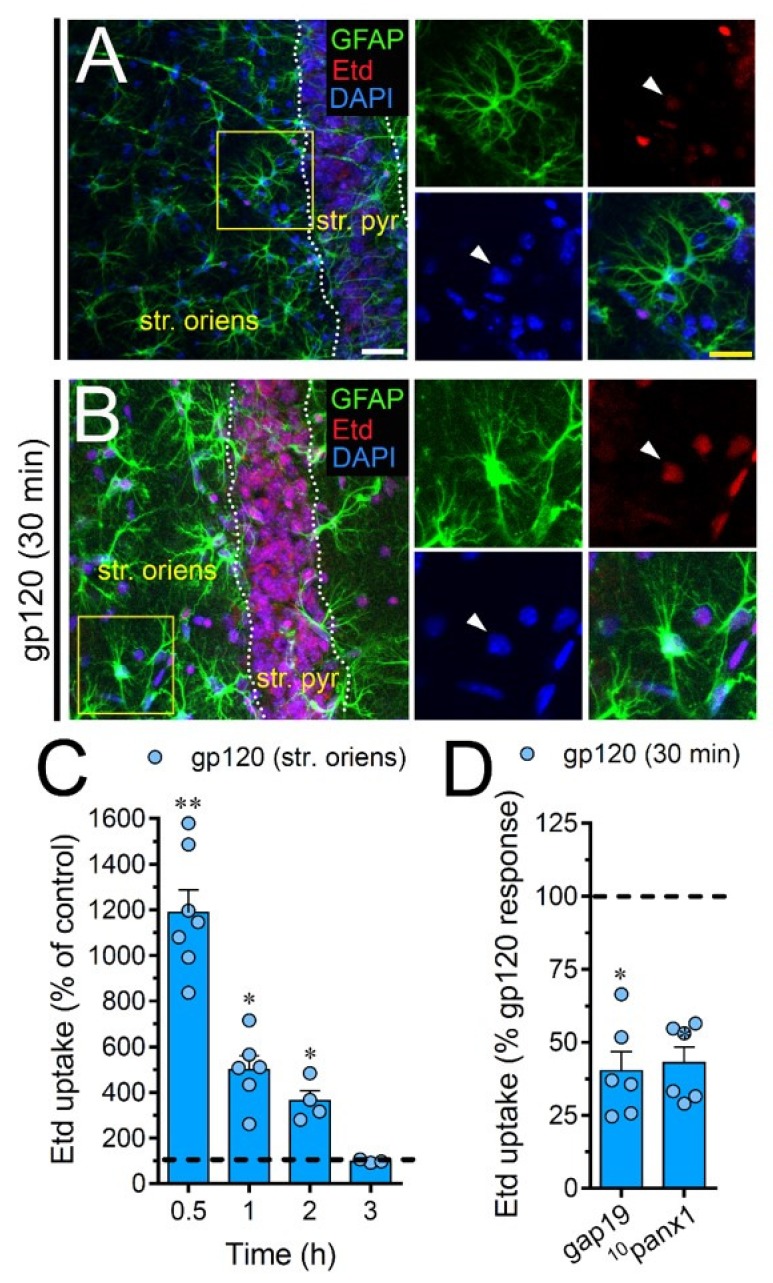
gp120 enhances Cx43 hemichannel and Panx1 channel function in hippocampal astrocytes from the stratum oriens. (**A**,**B**) GFAP (green), Etd (red) and DAPI (blue) staining in the stratum oriens of brain slices under control conditions (**A**) or stimulated for 30 min with 10 ng/mL gp120 (**B**). Insets of astrocytes were taken from the area depicted within the yellow squares in panels A and B, whereas arrowheads indicate their cell bodies. (**C**) Etd uptake normalized to control (dashed line) by astrocytes in the stratum oriens from brain slices after distinct periods of exposure to 10 ng/mL gp120. ** *p* < 0.001, * *p* = 0.05, gp120 vs. control (≥ 35 cells analyzed for at least three independent experiments). (**D**) Etd uptake normalized to the maximum response evoked by gp120 (dashed line) by astrocytes in the stratum oriens from brain slices after 30 min of exposure to 10 ng/mL plus gap19 (100 µM) or ^10^panx1 (100 µM). * *p* < 0.01, 30 min gp120 vs. blockers. (≥ 33 cells analyzed for at least three independent experiments). Calibration bars: white bar = 120 µm; yellow bar: 85 µm.

**Figure 8 ijms-21-02503-f008:**
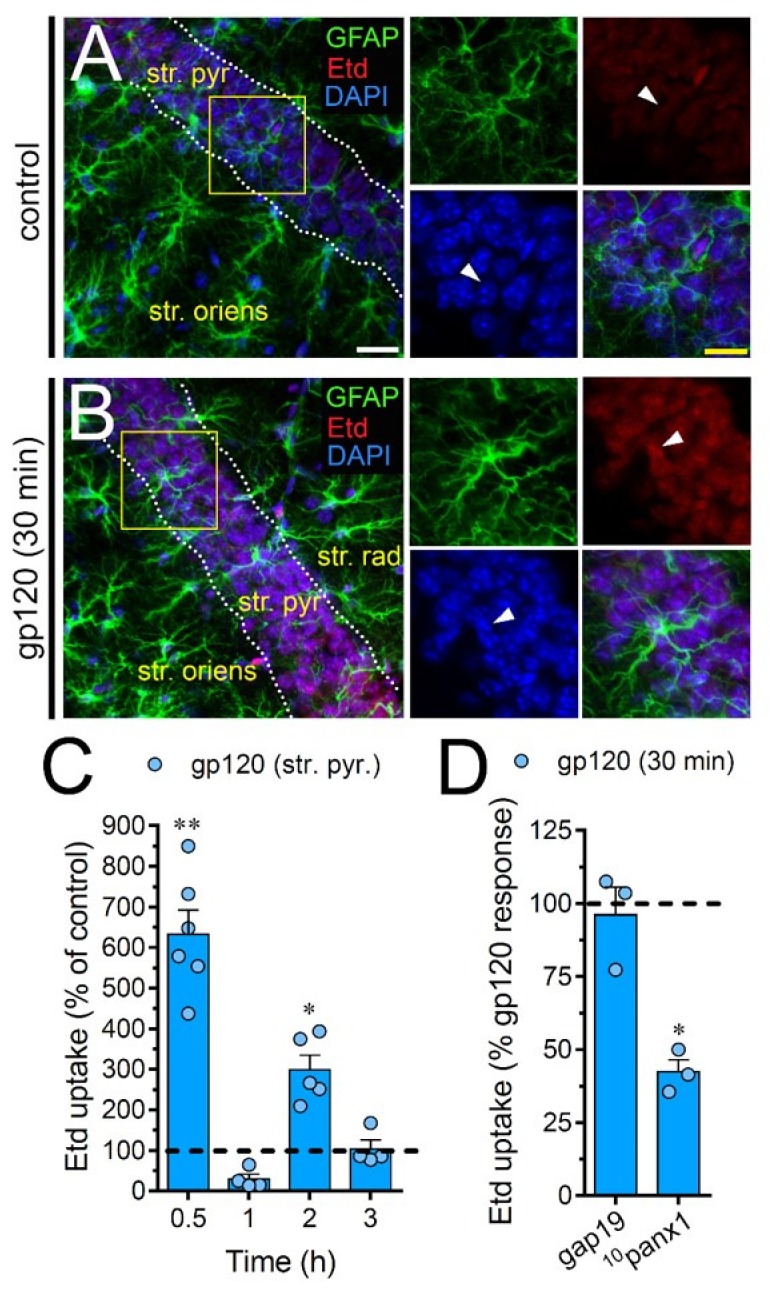
gp120 enhances Panx1 channel function in hippocampal astrocytes from the stratum pyramidale. (**A**–**B**) GFAP (green), Etd (red) and DAPI (blue) staining in the stratum pyramidale of brain slices under control conditions (**A**) or stimulated for 30 min with 10 ng/mL gp120 (**B**). Insets of astrocytes were taken from the area depicted within the yellow squares in panels A and B, whereas arrowheads indicate their cell bodies. (**C**) Etd uptake normalized to control (dashed line) by astrocytes in the stratum pyramidale from brain slices after distinct periods of exposure to 10 ng/mL gp120. ** *p* < 0.001, * *p* = 0.05, gp120 vs. control. (≥ 35 cells analyzed for at least three independent experiments). (**D**) Etd uptake normalized to maximum response evoked by gp120 (dashed line) by astrocytes in the stratum pyramidale from brain slices after 30 min of exposure to 10 ng/mL gp120 plus gap19 (100 µM) or ^10^panx1 (100 µM). * *p* < 0.01, 30 min gp120 vs. blockers (≥ 33 cells analyzed for at least three independent experiments). Calibration bars: white bar = 120 µm; yellow bar: 85 µm.

**Figure 9 ijms-21-02503-f009:**
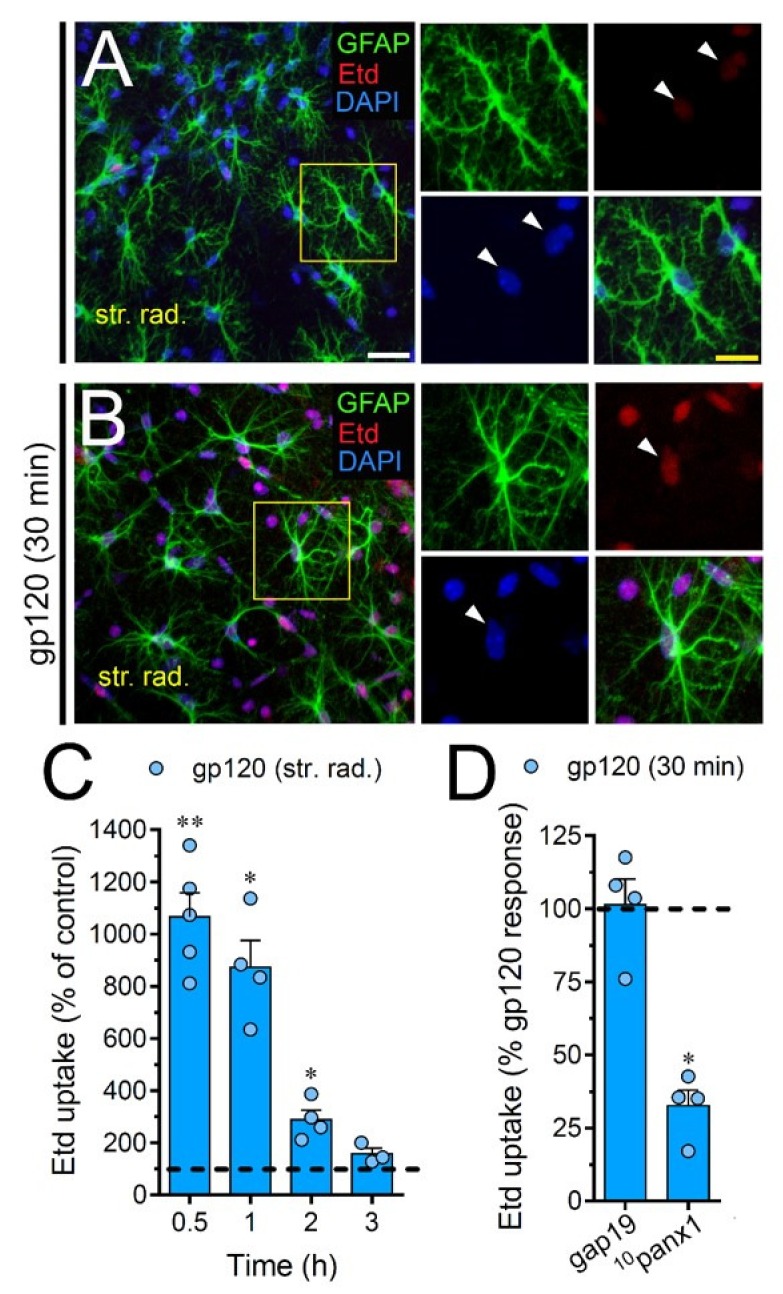
gp120 enhances Panx1 channel function in hippocampal astrocytes from the stratum radiatum. (**A**,**B**) GFAP (green), Etd (red) and DAPI (blue) staining in the stratum radiatum of brain slices under control conditions (**A**) or stimulated for 30 min with 10 ng/mL gp120 (**B**). Insets of astrocytes were taken from the area depicted within the yellow squares in panels A and B, whereas arrowheads indicate their cell bodies. (**C**) Etd uptake normalized to control (dashed line) by astrocytes in the stratum radiatum from brain slices after distinct periods of exposure to 10 ng/mL gp120. ** *p* < 0.001, * *p* = 0.05, gp120 vs. control (≥ 35 cells analyzed for at least three independent experiments). (**D**) Etd uptake normalized to the maximum response evoked by gp120 (dashed line) by astrocytes in the stratum radiatum from brain slices after 30 min of exposure to 10 ng/mL gp120 plus gap19 (100 µM) or ^10^panx1 (100 µM). * *p* < 0.01, 30 min gp120 vs. blockers (≥ 33 cells analyzed for at least three independent experiments). Calibration bars: white bar = 120 µm; yellow bar: 85 µm.

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
