# Peer review of "HIV gp120 Protein Increases the Function of Connexin 43 Hemichannels and Pannexin-1 Channels in Astrocytes: Repercussions on Astroglial Function"

_ijms, 2020, doi:10.3390/ijms21072503_

Round 1

Reviewer 1 Report

This manuscript demonstrates a role for the HIV protein gp120 in activation of astroglial Cx43 hemichannels and pannexin-1 channels that could provide insight into the pathogenesis of HIV-associated neurocognitive disorders. The study appears to be well performed and provides both in vitro and ex vivo data to support its conclusions. The only minor issue the reviewer has is that the manuscript requires English editing. A few examples of problems are given below. These and other minor edits are required.

Abstract

Of note, the gp120-induced channel opening of resulted in alterations in Ca2+ dynamics, nitric oxide production and ATP release.

Introduction

Although the antiretroviral therapy has 43 reduced the severity of these disorders in HIV-infected patients, the prevalence of HAND in these 44 individuals remains high (~50%) as they live longer and due the relatively poor BBB penetrance of 45 most antiretroviral drugs [11-14]. 

The wide-range CNS damage observed in HIV-infected individuals 46 with effective ART therapy had led to think that additional and novel mechanisms of bystander cell 47 damage could be implicated. 

Results

Consequently, we scrutinize the impact of these pathways in 139 the gp120-induced Cx43 hemichannel and Panx1 channel activity in astrocytes. 

Nevertheless, gap19 (100 μM) was successful in diminish Etd uptake only in the stratum oriens 308 (~33%, Fig. 7D) but not at the stratum pyramidale (Fig. 8D) and stratum radiatum (Fig. 9D). 

Discussion

Downstream signaling of 397 IL-1β/TNF-α and p38 MAPK lead to the expression of iNOS [77] and, in consequence, a raise in NO 398 production [78]. 

We conjecture that ATP release may serve as a downstream mechanism resulting in the aperture 414 of Cx43 hemichannel and/or Panx1 channels, 

Result of the use of acute brain slices, we 438 corroborate in a more integrated system the stimulatory influence of gp120 

Author Response

Reviewer 1.

This manuscript demonstrates a role for the HIV protein gp120 in activation of astroglial Cx43 hemichannels and pannexin-1 channels that could provide insight into the pathogenesis of HIV-associated neurocognitive disorders. The study appears to be well performed and provides both in vitro and ex vivo data to support its conclusions. The only minor issue the reviewer has is that the manuscript requires English editing. A few examples of problems are given below. These and other minor edits are required.

Response: We thank the reviewer for this comment. This revised version of our manuscript was edited by a well-known service for English Language Editing, which greatly improved our manuscript and reduced grammatical, spelling, and other common errors.

Abstract

Of note, the gp120-induced channel opening of resulted in alterations in Ca2+ dynamics, nitric oxide production and ATP release.

Response: We thank the reviewer for noting this mistake. In this revised version of our manuscript, we amended the sentence.

Introduction

Although the antiretroviral therapy has 43 reduced the severity of these disorders in HIV-infected patients, the prevalence of HAND in these 44 individuals remains high (~50%) as they live longer and due the relatively poor BBB penetrance of 45 most antiretroviral drugs [11-14].

Response: We thank the reviewer for noting this mistake. In this revised version of our manuscript, we amended the sentence.

The wide-range CNS damage observed in HIV-infected individuals 46 with effective ART therapy had led to think that additional and novel mechanisms of bystander cell 47 damage could be implicated.

Response: We thank the reviewer for noting this mistake. In this revised version of our manuscript, we amended the sentence

Results

Consequently, we scrutinize the impact of these pathways in 139 the gp120-induced Cx43 hemichannel and Panx1 channel activity in astrocytes.

Response: We thank the reviewer for noting this mistake. In this revised version of our manuscript, we amended the sentence

Nevertheless, gap19 (100 μM) was successful in diminish Etd uptake only in the stratum oriens 308 (~33%, Fig. 7D) but not at the stratum pyramidale (Fig. 8D) and stratum radiatum (Fig. 9D).

Response: We thank the reviewer for noting this mistake. In this revised version of our manuscript, we amended the sentence

Discussion

Downstream signaling of 397 IL-1β/TNF-α and p38 MAPK lead to the expression of iNOS [77] and, in consequence, a raise in NO 398 production [78].

Response: We thank the reviewer for noting this mistake. In this revised version of our manuscript, we amended the sentence

We conjecture that ATP release may serve as a downstream mechanism resulting in the aperture 414 of Cx43 hemichannel and/or Panx1 channels,

Response: We thank the reviewer for noting this mistake. In this revised version of our manuscript, we amended the sentence

Result of the use of acute brain slices, we 438 corroborate in a more integrated system the stimulatory influence of gp120

Response: We thank the reviewer for noting this mistake. In this revised version of our manuscript, we amended the sentence

Reviewer 2 Report

The intention of the study was to elucidate more in details the implication of connexin-43 hemichannels and pannexin-1 channels in damage of the brain of HIV-infected patients using soluble viral proteins gp120 and cultured mouse astrocytes. It has been demonstrated that gp120 increases the opening of both types of channels depending on the activation of IL-1β/TNF-α, p38 MAP kinase, iNOS, cytoplasmic Ca2+ and purinergic signaling. Moreover, gp120-induced channel opening resulted in alterations in Ca2+ dynamics, nitric oxide production and ATP release. Heterogeneous channel opening was reproduced in ex vivo brain preparations. These novel findings provide an important information related to prevention or attenuation neurocognitive disorders as well as reveal the targets for the treatment. Nevertheless, several points should be addressed before publication.

Title: Instead activity the function of channels is more appropriate term. It should be used throughout the manuscript.

Abstract should be factual avoiding words like however, nevertheless, of note, etc. Check please sentence in line 27 because of typing error.

Key words should include the most important navigation words (e.g. gp120, instead ATP, Cx43 hemichannels instead gap junctions).

Introduction:  Avoiding of long sentences is recommended. Findings should not be included in this Chapter.

Results: How did you explain dose-dependent effect of gp120 on Etd uptake (Fig 1.A)? Why gp120 was most effective in concentration of 10ng/ml? Shortening of Fig. 1 legend would be appreciated perhaps by elimination of statistic in this and all legends, because it is noted in Methods chapter.

Line 168, Perhaps it is misinterpreting that “Gap junctional-mediated coupling between astrocytes controls the extracellular concentrations of glutamate, K+ and H+,……

Line 179, instead granules Cx43 immunopositive spots

Findings associated with Fig 3, How did you recognize gap junction related Cx43 immunopositivity from Cx43 hemichannels related immunopositivity?

Line 384-385, Cx43 immunopositivity does not refer to the function of Cx43 channels.

Author Response

Reviewer 2

The intention of the study was to elucidate more in details the implication of connexin-43 hemichannels and pannexin-1 channels in damage of the brain of HIV-infected patients using soluble viral proteins gp120 and cultured mouse astrocytes. It has been demonstrated that gp120 increases the opening of both types of channels depending on the activation of IL-1β/TNF-α, p38 MAP kinase, iNOS, cytoplasmic Ca2+ and purinergic signaling. Moreover, gp120-induced channel opening resulted in alterations in Ca2+ dynamics, nitric oxide production and ATP release. Heterogeneous channel opening was reproduced in ex vivo brain preparations. These novel findings provide an important information related to prevention or attenuation neurocognitive disorders as well as reveal the targets for the treatment. Nevertheless, several points should be addressed before publication.

Title: Instead activity the function of channels is more appropriate term. It should be used throughout the manuscript.

Response: We thank the reviewer for this suggestion. In this revised version of our manuscript, we replaced the term “activity” by “function” throughout the manuscript.

Abstract should be factual avoiding words like however, nevertheless, of note, etc. Check please sentence in line 27 because of typing error.

Response: We thank the reviewer for these suggestions. In this revised version of our manuscript, we amended these sentences.

Key words should include the most important navigation words (e.g. gp120, instead ATP, Cx43 hemichannels instead gap junctions).

Response: We thank the reviewer for this suggestion. In this revised version of our manuscript, we amended the key words.

Introduction:  Avoiding of long sentences is recommended. Findings should not be included in this Chapter.

Response: We thank the reviewer for this suggestion. In this revised version of our manuscript, we reduce the word count in several sentences in the introductory section.

Results: How did you explain dose-dependent effect of gp120 on Etd uptake (Fig 1.A)? Why gp120 was most effective in concentration of 10ng/ml? Shortening of Fig. 1 legend would be appreciated perhaps by elimination of statistic in this and all legends, because it is noted in Methods chapter.

Response: We thank the reviewer for this comment. One explanation may be related with the fact that several inflammatory responses involving cytokine signaling occur with a bell-shaped manner. The latter because cytokines have complex and multiple downstream targets (PMID: 24388212). Alternatively, most of P2Y receptors undergo desensitization with high concentrations or chronic exposure to agonists (PMID: 26519900).

In this revised version of our manuscript, we eliminate part of the statistic already mentioned in the method section.

Line 168, Perhaps it is misinterpreting that “Gap junctional-mediated coupling between astrocytes controls the extracellular concentrations of glutamate, K+ and H+,……

Response: We thank the reviewer for this comment. In this revised version of our manuscript, we amended the sentence in order to avoid any confusion.

Line 179, instead granules Cx43 immunopositive spots

Response: We thank the reviewer for these suggestions. In this revised version of our manuscript, we amended this sentence.

Findings associated with Fig 3, How did you recognize gap junction related Cx43 immunopositivity from Cx43 hemichannels related immunopositivity?

Response: We thank the reviewer for this comment. In theory, using immunofluorescence analysis we cannot discriminate between the Cx43 that is forming hemichannels vs gap junctions. However, the attempt of Figure 3 is to give a qualitative but not quantitative inspection of Cx43 in plaque-like structures at cell-cell interfaces. Moreover, it has been shown that surface hemichannels account for ∼11% of total Cx43 under resting conditions (Schalper et al., 2008. Cell Commun Adhes. 15(1):207-18), making them poorly detectable by immunofluorescence. Thereof, changes in Cx43 immunodetection do not necessarily implicate changes in hemichannel activity. Indeed, the evidence has shown that increased opening of astroglial Cx43 hemichannels occurs despite a reduction in Cx43 immunodetection under various conditions, including: cytokine mediated inflammation (Retamal et al., 2007. J Neurosci), Niemann-Pick Type C disease (Sáez et al., 2013.Plos One) and neuronal ceroid lipofuscinosis (Burkovetskaya et al., 2014.Plos one).

Line 384-385, Cx43 immunopositivity does not refer to the function of Cx43 channels.

Response: We thank the reviewer for this comment. In this revised version of our manuscript, we amended this sentence.